# A Comprehensive Survey on Certificate-Less Authentication Schemes for Vehicular Ad hoc Networks in Intelligent Transportation Systems

**DOI:** 10.3390/s23052682

**Published:** 2023-03-01

**Authors:** Santhosh Kumar Sripathi Venkata Naga, Rajkumar Yesuraj, Selvi Munuswamy, Kannan Arputharaj

**Affiliations:** 1School of Information Technology and Engineering, Vellore Institute of Technology, Vellore 632013, Tamilnadu, India; 2School of Computer Science and Engineering, Vellore Institute of Technology, Vellore 632014, Tamil Nadu, India

**Keywords:** smart cities, intelligent transportation systems, certificate-less authentication schemes, vehicular ad hoc networks, anonymity

## Abstract

Data transmission in intelligent transportation systems is being challenged by a variety of factors, such as open wireless communication channels, that pose problems related to security, anonymity, and privacy. To achieve secure data transmission, several authentication schemes are proposed by various researchers. The most predominant schemes are based on identity-based and public-key cryptography techniques. Due to limitations such as key escrow in identity-based cryptography and certificate management in public-key cryptography, certificate-less authentication schemes arrived to counter these challenges. This paper presents a comprehensive survey on the classification of various types of certificate-less authentication schemes and their features. The schemes are classified based on their type of authentication, the techniques used, the attacks they address, and their security requirements. This survey highlights the performance comparison of various authentication schemes and presents the gaps in them, thereby providing insights for the realization of intelligent transportation systems.

## 1. Introduction

Intelligent Transportation Systems (ITS) have made promising advancements in addressing the problems related to pollution and traffic management [1]. The emergence of the Internet of Things (IoT) has been revolutionary, with a variety of applications that benefit human lives, business operations, transactions, parking, governmental organizations, and so on. It consists of wide variety of sensors that capture data by sensing, and it deploys the information to the base station as a backup for analysis and analytics to gain business and research insights [2]. ITS can be regarded as a kind of heterogeneous network which comprises intra- and inter-vehicular communications. The most predominant application of the IoT is vehicular ad hoc networks (VANETs). They have captured huge attention from many researchers all around the world [3,4]. VANETs consist of cars equipped with On-board Units (OBUs) which facilitate communication via a dedicated short-range communication protocol (DSRC) [5]. Sensors installed inside the cars sense the environment and issue a collision warning in cases of vehicle-to-vehicle communications (V2V). These kinds of communications can be utilized for warning/alerting the drivers, thereby providing driving assistance. VANETs also consist of Road-side Units (RSUs), which are mainly necessary to provide service assistance to drivers in cases of accidents or traffic, thereby making commuters plan an alternative route in advance [6]. This kind of infrastructure communication can be referred as vehicle-to-infrastructure communication (V2I). The collected information has to be stored in a secured environment. If there is no proper security infrastructure to guard the data, then they may be prone to attacks. This might pose the attackers to even hack the IoT device (cars), which increases the security risk. Hence, security and privacy have been under threat due to new methods of communication and hacking methods. Figure 1 represents the typical components of Intelligent Transportation Systems in a smart city.

### 1.1. Problem Statement

Since VANETs are resource-constrained, they pose challenges, as they have less computational resources. Due to wide-open communication channels in recent years, end–end security has been a question in need of being addressed. Before a message is transferred, the data have to be owned and encrypted to emphasize that they originate from an authorized device and that they can stand alone in cases of attacks during transmission. The data transferred might contain important or highly sensitive information that should not be tampered with or leaked. In Vehicular Ad hoc Networks (VANETs), when a large number of On-board Units send data to a particular Road-side Unit, then it might take time to verify the integrity and authenticity of the data that are received. This leads to a processing delay, where the Road-side Units might become tired and announce a denial-of-service. This delay, in the case of sensitive information, may pose a huge risk. Additionally, they are easily exposed to wear and tear, ease of access, physical threats, and side-channel and cloning attacks [7]. Due to the high mobility and streamlining of resources, it is very difficult to interpret the nodes that behave malevolently. It is also possible that these malevolent nodes may send or broadcast wrong data to other nodes and might cause disturbances in the routing and data transmission. VANETs suffer from a wide variety of attacks, including message tampering, message spoofing, denial-of-service (DoS), and so on [8]. In order to ensure the security of VANETs, the design has to include principles such as authenticity, privacy, and integrity.

### 1.2. Background

Authenticity can be termed as ensuring the credibility of the sender to perform secure routing. Privacy refers to the protection of the information against third parties; this can be achieved using pseudonyms. Integrity is the mechanism of ensuring that the data are tamper-proof, and not altered or lost during transmission. Public-key cryptography (PKG) comes up with a solution termed as Digital Signature [9,10]. Digital Signature provides facilities for signing each message so that it ensures authentication, integrity, and confidentiality. In this approach, data transmission by a sender is acknowledged first by the receiver’s public key. If the sender has the recipient’s public key, then the sender authenticates the message by using the received public key. During transmission, the Road-side Units (RSUs) and Traffic Regulating Authority (TRAs) perform Signature Verification. Signature Verification takes a long processing time, which increases the delay and computational overhead in the cases of dense and sparse scenarios [11,12]. In order to solve this challenge, Shamir et al. [13] proposed a cryptographic technique called identity-based public-key cryptography (ID-PKC). In this approach, a public key can be generated by the user themselves, whereas the corresponding private key is generated by the Key Generation Center (KGC). Though it solves the problem of the management of certificates with public-key cryptography, it still suffers from a serious challenge. The Key Generation Center contains all the users’ private keys. At any time, if the Key Generation Center is attacked and captured, then the attacker is able to interpret the data that are to be communicated. The forging of signatures may also be possible, which poses a high risk. This forging of signatures attack is called a key escrow problem. In order to solve the deficiencies of the two cryptosystems, Al-Riyami et al. [14] came up with a solution termed as certificate-less public-key cryptography (CL-PKC). In this approach, the user’s private key is generated by the Key Generation Center. However, only part of the key can be generated by the KGC, and the remaining part has to be generated by the user. Thus, even if the attacker captures the KGC, it is impossible for them to gain access to the private information. Additionally, it eliminates the need for the management of certificates. Thus, both the certificate management and key escrow problems are solved. However, this still poses an overhead of a large number of certificates to be processed. To eliminate this aggregate, signature technology, along with the certificate-less public-key cryptography, provides a better solution. Instead of processing the individual signatures, which increases the computational overhead in a resource-constrained environment like VANETs, all these signatures are aggregated into a single signature. This kind of approach reduces the processing time and also improves the verification efficiency [15]. Therefore, certificate-less public-key cryptography, along with aggregate signature technology, has attracted the attention of many researchers around the world. Based on the ways in which the signature is authenticated, the certificate-less aggregate schemes (CLAS) are classified into different categories, namely batch verification, group verification, mutual authentication, cooperative authentication, hybrid authentication, and others. Figure 2 represents the taxonomy of the authentication schemes utilized for vehicular ad hoc networks.

### 1.3. Our Contributions

This paper aims to provide a comprehensive survey on the various types of certificate-less public-key cryptography-based authentication schemes. An analysis has been carried out and presented with existing surveys that are highlighted in Figure 2. From the analysis carried out, the significance of the survey has been constructed, as shown in Table 1.

### 1.4. Approach

#### 1.4.1. Survey Organization

The survey has been carried out after a careful selection of more than 200 papers pertaining to the topic of certificate-less authentication schemes. To resolve the conclusion, the papers are segregated based on the signature verification technique applied, and the parameters used that address various issues. We then confined our scavenging with a case study that categorized based on the signature verification technique, as depicted in the Figure 3. The classification has been performed based on the Figure 4 as shown below.

#### 1.4.2. Queries

Research questions pertaining to the proposed survey have been analyzed and identified, refining the process of conducting the literature survey. Table 2 defines the set of queries that confined the way for the proposed survey.

#### 1.4.3. Literature Sources

The literature survey has been backed up from both open access and peer-reviewed publications from the following list of journal databases that includes:➢ACM (https://www.acm.org)➢Wiley (https://www.wiley.com/en-us)➢Taylor and Francis (https://www.tandfonline.com)➢Open Access (https://www.mdpi.com)➢IEEE Xplore (ieeexplore.ieee.org)➢Springer (link.springer.com)➢ScienceDirect (www.sciencedirect.com)

The publications have been selected carefully, covering papers from the year 2012 to 2022, out of which most of them are from 2022. 

#### 1.4.4. Notion of Search

During the scavenging of literature sources, the keywords “Authentication schemes”, “Vanets”, “Survey”, and “Year” were the main words utilized for the hunt. In order to confine and to conclude within the domain, keywords such as “security” and “privacy” were also utilized.

This survey enables future researchers to gain an insight into designing an efficient cryptography scheme for VANETs in Intelligent Transportation Systems (ITS). The manuscript has been organized as follows: Section 1 describes the problem statement with an introduction. Section 2 provides a brief on the related works carried out by various researchers pertaining to certificate-less aggregate schemes. Section 3 explains the performance comparison of different certificate-less aggregate signature schemes, along with their gaps. Section 4 provides the conclusion and future perspectives.

**Table 1 sensors-23-02682-t001:** Comparison with existing surveys.

Author	Year	Goal	Requirement	Advantages	Limitations
Zhaojun Lu et al. [16]	2018	Compact Survey on anonymous authentication schemes under pseudonyms	To focus on novel privacy-preserving methods and trust models by filling gaps and to report the recent advancements	Provides clarity on privacy preservation and trust model design	Lacks clarity in addressing the issues pertaining to different authentication schemes
Shaik Mullapathi Farooq et al. [17]	2021	Surveyed on different authentication mechanism in VANETs	To provide a clarity on security attacks (both inside and outside)	Provides research directions and arena for security mechanisms to be implemented	Lacks in addressing verification methods addressed in different authentication schemes
Eko Fajar Cahyadi et al. [18]	2022	Presented a survey on identity-based batch verification schemes	To provide a review on identity-based batch verification schemes	Very limited but clarity in presenting the approach	The major drawback is that it has not considered other verification mechanisms
J.Jenefa et al. [19]	2022	Presented a survey on existing authentication schemes	To provide a survey based on message signing and verification methods	Provides enough clarity and classified different authentication methods in a precise way	It has not addressed the classification based on wide collection of literature sources
Proposed	2022	Presenting a survey based on verification methods and various authentication schemes	To provide a clarified survey on different techniques, parameters, methods, and verification schemes	Provides a research direction that address the future challenge	-

**Table 2 sensors-23-02682-t002:** Queries.

Queries	Aim
1. What are Intelligent Transportation Systems?	They consist of wide variety of sensors that capture data by sensing and deploying the information to the base station as a backup for analysis and analytics of efficient traffic management and transportation.
2. What are VANETs?	VANET consists of cars equipped with On-board Units (OBUs), facilitating communication via dedicated short-range communication protocols (DSRC).
3. What are issues faced by VANETs?	Security, privacy, delay, and bandwidth are the issues faced due to the varying dynamic topology nature.
4. Why do VANETs face security and privacy issues?	Due to the wide-open nature of communication channels, VANETs are highly subjected to security and privacy risks.
5. How to face security and privacy issues?	Data has to be authenticated prior to transmission. Due to open communication channels, it is necessary to devise an authentication scheme that guards the data being transferred between VANETs.
6. What is an authentication scheme for VANETs?	It is the way of acknowledging the user/sender identity, thereby gaining access to the resource or the message via a password, pseudonym, and so on.
7. How are they classified?	They are classified based on certificates, without certificates and authentication mechanisms such as mutual, cooperative, hybrid, and other types.
8. What is an attractive feature of authentication schemes based on certificates and without certificates?	In the case of certificate authentication schemes, there is no attraction because of the overhead incurred in the processing, managing, and verification of them. In the case of certificate-less authentication schemes, it alleviates the certificate processing management overhead.
9. How are certificate-less authentication schemes classified?	They are classified based on the verification strategy and the authentication scheme employed.

## 2. Taxonomy of Classification of Different Certificate-Less Authentication Schemes

Authentication in VANETs can be achieved in two ways: authenticating infrastructures such as vehicles and messages generated. Infrastructure authentication is vital to ensuring the authenticity of the entity deployed inside the network. It is also necessary to ensuring whether the legitimacy of the entity has been compromised by the attacker or not. Message authentication is necessary to ensure the integrity of the data. Various authors have proposed a lot of research work pertaining to different security requirements in certificate-less aggregate schemes (CLASs) to achieve security and privacy. Figure 5 illustrates the classification of different authentication mechanisms based on message signature verification techniques. 

### 2.1. Classification Based on Signature Verification Technique 

Based on the verification mechanism followed, certificate-less aggregate signature schemes are classified into batch verification and group verification methods.

#### 2.1.1. Batch Authentication

Digital Signatures can be verified or authenticated by batch. Batch verification can be taken as an approach to verify a collection of individual signatures at the same time. It is essential to reduce the computation time spent on verifying the signature rather than communication [20]. The following are the works carried out by various authors utilizing the batch verification process.

Felix Vallant et al. [21] proposed a CLAS scheme for VANETs in a smart grid environment. The major aim of the work is to achieve user anonymity, along with conditional privacy protection. The advantage of this scheme is that expensive operations such as bilinear pairings and Map-to-Point hash functions are not used. Though it reduces the computation and communication costs, the efficiency is on par with the other related works, which need to be improved further. Ismaila Adeniyi Kamil et al. [22] came up with a solution by finding and mitigating the security vulnerability in Cui et al. [23] by addressing the forgery under chosen messages. The advantage of the proposed work is that expensive operations like bilinear pairings are not used. A separate verifier is used for verifying the signatures, irrespective of the number of signatures used. The scheme can be utilized for several practical applications pertaining to safety, efficacy, infotainment, and user comfort ITS applications. Cui et al. [23] addressed privacy protection and key escrow problems. The major aim of their work was to achieve privacy preservation. One-way Hash is used for batch verification. The scheme mitigates forgery attack. Huei-Ru Tseng et al. [24] addressed the issue of the security and privacy of emergency or sensitive message dissemination. The scheme was proposed by them to defend against the forgery, privacy, and traceability of vehicles during data transmission. Bilinear pairing is a technique utilized for implementing security. This scheme does not support vehicle-to-infrastructure (V2I) communications. Samra et al. [25] came up with a ring-signature-based method to address the problem of privacy. A certificate-less ring signature scheme with batch verification was proposed by the authors. The major aim of their work is to preserve or protect privacy, authentication, integrity, and traceability. Their scheme uses a One-way Hash function for implementing security. However, ring signatures suffer a typical computation overhead. Guo et al. [26] proposed a different methodology based on a ring encryption scheme to address conditional privacy protection. The scheme supports both vehicle-to-vehicle (V2V) and V2I communications. The scheme mitigates the replay attack. However, it passively resists insider attacks. Additionally, the scheme has to be tested for urban scenarios. Ye et al. [27] addressed the problem of coalition attacks with the help of pseudonyms to achieve anonymity. Their proposed scheme mitigates forgery against both chosen messages and coalition attacks. The scheme suffers from a low efficacy due to malicious signatures during the aggregation. Wang et al. [28] addressed the problems of privacy in vehicles, messages, and other bandwidth-related constraints. This scheme mitigates forgery and replay attacks. Their proposed work has to be tested under real-time constraints. Chen et al. [29] addressed the problems of vehicle identity privacy, location privacy, high mobility, and limited bandwidth. Their scheme was proposed with the objective of achieving conditional privacy protection. The scheme consists of a message authentication protocol which has been enabled to counter forgery under chosen messages. Han et al. [30] developed a pairing-free certificate-less aggregate signature scheme to improve the security in vehicle-to-infrastructure communication. The scheme was proposed to address the problem of limited bandwidth and computing power in the case of VANETs. This scheme mitigates forgery attacks. The efficiency of their proposed scheme has to be improved further. Thumbur et al. [31] addressed the problem of message modification and the misuse of private information. The proposed scheme works for both V2V and V2I communications. The major advantage of it is that it means expensive operations like bilinear pairing are not used. It prevents malicious vehicles from disrupting the security features. Gayathri et al. [32] proposed a scheme to address the problems of security and privacy for healthcare-based wireless medical sensor networks. Their proposed scheme effectively resists signature forgery attacks. Expensive operations like bilinear pairings and Map-to-Point Hash functions are not used. Though their work achieves practicability, it has to be tested by designing novel attacks. Zhou et al. [33] proposed a scheme to address the problems of security and privacy by mitigating against signature forgery attacks. The scheme has to be tested by novel attacks so that it reveals its suitability for a particular scenario. Wang et al. [34] addressed the problem of security over wide-open communication channels for federated UAV networks. Their proposed work utilizes a certificate-less and fuzzy-based batch authentication scheme that aims to protect against forgery upon chosen message attacks. Wang et al. [35] came up with this by finding a flaw or secure vulnerability in the work of Li et al. [35]. The work performed by Li et al. is insecure against unlinkability and anonymity. The scheme utilized a random number to encrypt the unique tag in order to ensure anonymity. The scheme still suffers from computational overhead, which needs to be investigated further. Sunday et al. [36] proposed a lightweight and privacy-preserving certificate-less authentication scheme in fog-assisted VANETs, using blockchain and neuro-fuzzy machine learning techniques. The major aim of their scheme was to achieve a decentralized transparent and revocation process, and to prevent DoS attacks. Their proposed scheme mitigates forgery, replay, modification, impersonation, and man-in-the-middle attacks. The scheme has to be improved further to tackle novel types of attacks. Yang et al. [37] came up with a privacy-preserving aggregation authentication scheme for safety warning systems in fog-cloud based VANETs. Their proposed scheme utilizes bilinear maps for security implementation, thereby mitigating replay and forgery attacks. The limitation of their scheme is that the computation cost has to be reduced further. Ping et al. [38] proposed a pairing-free certificate-less aggregation authentication scheme to address the key escrow problem. The proposed scheme poses much effect against chosen message and identity attacks. Li et al. [39] came up with a solution to the flaw in the work proposed under the computational Diffie–Hellman assumption (CDH). The work is insecure against message signature forgery attacks. Hence, the proposed solution mitigated the flaw by making an improvement; hence, the scheme becomes superior against such attacks. Liu et al. [40] proposed a solution for Road-side Units (RSUs) due to the difficulty of verifying message signatures. The scheme works based on a proxy-vehicle to assist the RSU. In order to facilitate the data transmission, an expedite key negotiation scheme has been utilized. Their proposed work showed a significant improvement over the existing batch-based authentication schemes. However, it is crucial to make sure that these vehicles have incentives to serve others under the conditions of efficient message delivery. Another important challenge of the proposed scheme is that the redundant authentication event that happens frequently has to be addressed. Sai Ji et al. [41] proposed a scheme to address the problems of security and privacy in wireless body area networks (WBAN), utilizing big data services. The proposed scheme has tried to mitigate replay, impersonation, modification, a lost personal digital assistant (PDA), and man-in-the-middle (MitM) attacks. The proposed scheme was not tested under real-time scenarios. Zhong et al. [42] proposed a privacy-preserving authentication scheme with full aggregation. Their proposed work addresses the problems of latency and the integrity of data. The work involves protecting the identity and privacy of the mobile devices, and the attacks can be traced by using the pseudonym. The secure communication between a vehicle and multiple mobile devices is realized by certificate-less ring signature technology. However, security threat models need to be tested for various attacks. Hu et al. [43] addressed the problems of forgery, traceability, and blindness. The authors introduced a two-level dummy for enhancing the security. Even if one of the dummies gets exposed to the attacks, the other dummy entity will increase the security by protecting the real identity of the vehicle. Xie et al. [44] proposed a conditional privacy-preserving secure certificate-less authentication scheme for wireless body area networks. Their proposed work has tried to achieve forgery and replay attacks. The flaw mitigated the adversarial effect on the loopholes pertaining to message forgery and redundant message events. Further investigation of the scheme has to be tested by designing innovative attack scenarios. Song et al. [45] proposed a pairing-free lightweight batch anonymous authentication scheme that supports both V2V and V2I communications. The main aim of their work is to eliminate central authority dependency. The major advantage of the scheme is that no bilinear pairing operations have been performed, which increases the computation and the storage overhead. Zhang et al. [46] utilized the Chinese remainder theorem (CRT) that eliminates the need to store the sensitive information in the tamper-proof devices (TPD). Though the work achieves practicability, its efficiency has to be improved further. Table 3 depicts the classification of batch authentication schemes along with the problems addressed, the methodology, and the scenarios they suit.

**Table 3 sensors-23-02682-t003:** Classification of batch-verification-based certificate-less aggregate signature schemes.

Schemes	Methodology	Requirement	Attack Resistivity	Scenario
Felix Vallant et al. [21]	Elliptic curves; One-way Hash function	User anonymity, conditional privacy	Impersonation, replay, modification, man-in-the-middle, and stolen-verifier table attacks	V2V, V2I, V2G, V2E, IoT, Smart Grid
Cui et al. [23]	Elliptic curves; Map-to-Point Hash function	Identity privacy preservation	Impersonation, replay, modification, man-in-the-middle and stolen-verifier table attacks	V2I
Huei-Ru Tseng et al. [24]	Bilinear pairing	Traceability, conditional privacy	Forgery attack	V2V
Samra et al. [25]	Bilinear pairing, ring signature	Unforgeability, anonymity, traceability, conditional privacy	Forgery attack	V2V, V2I
Guo et al. [26]	Elliptic curves, ring signature	Conditional privacy	Replay attack	V2V, V2I
Ye et al. [27]	Elliptic curves	Anonymity, conditional privacy	Forgery, adaptive chosen message, coalition attacks	IoV
Wang et al. [28]	Bilinear pairing	Integrity, non-repudiation, anonymity, unlinkability, traceability,	Forgery, chosen message, replay attacks	V2V, V2I
Han et al. [29]	Elliptic curves	Conditional privacy	Replay, impersonation, modification, message spoofing attacks	V2I
Zhou et al. [30]	Elliptic curves	Conditional privacy, anonymity, traceability	Signature forgery, replay, modification, man-in-the-middle attacks	V2V, V2I
Sunday et al. [36]	Elliptic curves; neuro fuzzy and blockchain	Privacy preservation	DoS Attacks, signature forgery, replay, message modification, man-in-the-middle attacks	V2V, V2I, V2E
Liu et al. [40]	Bilinear pairing, proxy vehicle, expedite key negotiation scheme	Message integrity, authentication, identity privacy preservation, traceability	Signature replay attacks	V2V, V2I
Zhong et al. [42]	Bilinear pairing, ring signature	Latency, authentication, revocation, unforgeability, integrity, identity privacy preservation, traceability	Signature forgery attacks	Connected Autonomous Vehicles (CAVs), Intelligent Connected Vehicles (ICVs)
Zhang et al. [46]	Elliptic curves, Chinese remainder theorem (CRT)	Conditional privacy preservation, identity privacy preservation, perfect backward secrecy, perfect forward secrecy, traceability	Impersonation, message modification, replay, collusion attacks	V2V, V2I

#### 2.1.2. Group Verification

Digital signatures are the integral part of authenticating the originality of data. Hence, group signature is an approach which facilitates each member of a group to sign a message while the verification ensures that the message is from a particular group, while the signing authority is not known [47]. The following are the works carried out by various authors based on the group authentication process. 

Tan et al. [48] addressed the problem of DoS attacks by eradicating the authentication of a large number of anomaly messages. They employed the Chinese remainder theorem to perform the group verification and updating, thereby eliminating the need for the wastage of the storage of tamper-proof devices. This scheme assists in the anomaly detection of Road-side Units. The work suffers from the limitation that it only considers the traffic information, but it does not take into account the traffic information during the data transfer from the vehicle. Ikram Ali et al. [49] proposed a blockchain-based certificate-less group authentication scheme to achieve conditional privacy and revocation transparency. The Chinese Remainder Theorem was used for group key distribution and updating, and the dynamic time warping algorithm was used for unsupervised anomaly detection. Though the work proposed mitigated the drawbacks of security and privacy threats, it is not suitable for V2I environments. Zhiyan Xu et al. [50] proposed a certificate-less aggregate signature scheme for classical issues such as certificate management and key escrow problems. RSUs and traffic centers in VANETs (generally verified by trusted authority) need to verify a large number of route-related signatures in high-density communication scenarios. The scheme is substantially more efficient than the other schemes. Wang et al. [51] carried out an analysis on the security vulnerabilities in the identity traceability and user identity privacy, and found that it is prone to attacks. The work is based on RSU-based authentication. It is not suited for V2V communications.

Ming et al. [52] addressed the problems of message recovery, certificate management, and key escrow. However, a performance investigation while using bilinear pairings and Map-to-Point Hash functions is to be used to determine its suitability. Zheng et al. [53] addressed the problem of anonymity within eavesdropping, positioning, and tracking a vehicle. Adding, signing, verifying, and the revocation of group members using the simple multiplication of elliptic curves has been used. The scheme saves much computation time due to partial key generation. A high computation overhead with certificate distribution and revocation, a strong resilience of tamper-proof devices, a limited scalability when building many secure channels, and an inability to detect between tampering attacks are some of the problems addressed by Li et al. [54]. Identity-based public key certificate-less authentication schemes are utilized to achieve privacy, anonymity, and traceability. The RSU overhead has been reduced. The computation overhead has been reduced to an extent. Ismaila et al. [55] found a flaw in Tan et al.’s [48] scheme, discovering that it could not prevent against signature forgery attacks. The work involves a lightweight certificate-less authentication scheme for information dissemination concerning the privacy of users. Gaya et al. [56] utilized a signcryption technique as an ideal way of disseminating messages in a logical, secure, and authenticated way. However, a real-time scenario investigation is yet to be conducted. Yang et al. [57] addressed the problems of certificate management and key escrow. The scheme is effective against attacks and satisfies the unforgeability, traceability, and anonymity of communication messages. The proposed scheme has to be tested for high density scenarios for its effectiveness. Zhong et al. [58] came up with a solution to address the issues of privacy preservation, computation overhead, and security. The proposed scheme achieves message authentication and saves bandwidth and computation resources. The work mitigates replay attacks. Though it achieves its practicability, still-improved lightweight schemes are needed. Zhao et al. [59] addressed the problems of key escrow and key management. Their scheme is suitable for V2I communications in urban scenarios. The schemes resist attacks and satisfy the unforgeability of the communication messages, as well as the traceability and anonymity of the vehicles. High-density scenarios are to be tested.

Hu et al. [60] investigated the flaws to defend against signature forgery attacks. The problems need to address limited energy and location privacy. Hu et al. [61] investigated the security flaw in the Teng and Wang scheme [48] and mitigated forgery attacks. Hu et al. [62] investigated and mitigated the forgery attacks. Tomar et al. [63] extensively carried out research to address the problems pertaining to certificate revocation list management, key distribution, key escrow, and high computation overhead. The authors studied the work of Cui et al. [23] and identified a flaw with the employment of multiple semi-trusted authorities. In their model, there was no cooperation between the multiple trusted authorities employed. Hence, in order to mitigate the issue, blockchain technology is used in order to achieve data security, thereby preventing data leakage attacks. The installation cost is going to soar high during implementation. Jun et al. [64] addressed the security and privacy at the regions where the Road-side Units were unavailable. Their proposed work needs to resist against known attacks with a lesser computation cost and to reduce the workload of the vehicles while performing verification. A lightweight certificate-less key agreement protocol without pairing has been proposed to counter the issue. The scheme does not suit V2I environments. Parthiban et al. [65] addressed the problem of user privacy through a certificate-less aggregate signature-based authentication scheme (CSBA). Further improvements are needed to address the challenges encountered during data transfer. The scheme does not suit vehicle-to-infrastructure environments. Yang et al. [66] addressed the problems of privacy, integrity, and non-repudiation. The work includes finding a flaw in Thumbur et al. [31], which is insecure against public key replacement and coalition attacks. The algorithm suffers from identity revocation and authentication is not considered in the scheme proposed.

Dewangan et al. [67] proposed a solution for side-channel attacks on OBUs. Due to classical cryptography problems such as certificate management and key escrow, side-channel attacks are more prevalent. Their proposed scheme is resistant to side-channel attacks and mitigates message forgeability behavior. However, the correctness of the proofs for traceability needs to be investigated. Pairat Thorncharoensri et al. [68] addressed security and privacy issues while sharing information over the cloud. This reduced the computation and the communication cost. But it still suffers from some of the issues that were solved by the benchmark Cui et al. [23]. Chattaraj et al. [69] proposed a new blockchain-based certificate-less key agreement protocol for the Internet of Vehicles (IoV). Elliptic curves have been used for implementing the security. Their proposed work successfully mitigated the pitfalls with the help of a blockchain. Mei et al. [70] addressed the problem of vehicle location privacy, and the authenticity in the case of Internet of Vehicles. The scheme has the ability to resist attacks from outside attackers and the malicious-but-passive KGC. The aggregate verification cost can be improved further. Table 4 provides the classification of various group authentication schemes along with the problem requirements, the methodology, and the scenario they suit.

**Table 4 sensors-23-02682-t004:** Classification of group-verification-based certificate-less aggregate signature schemes.

Schemes	Methodology	Requirement	Attack Resistivity	Scenario
Haowen Tan et al. [48]	Elliptic curves, Chinese remainder theorem (CRT), dynamic time warping algorithm (DTW)	Unsupervised anomaly detection	DoS, anomaly attacks	V2V, V2I
Ikram Ali et al. [49]	Bilinear pairing, blockchain	Conditional privacy, source authentication, identity privacy preservation, traceability, revocation transparency	Collusion, impersonation, message modification, man-in-the-middle, replay attacks	V2I
Zhiyan Xu et al. [50]	Bilinear maps	Privacy preservation	Forgery attack	V2V, V2I
Ming et al. [52]	Elliptic curves	Conditional privacy preservation, identity privacy preservation, traceability, role separation	Replay, modification, impersonation, man-in-the-middle attacks	V2I
Li et al. [54]	Elliptic curves	Traceability, impersonation	Chosen message, node compromise, node replication, stolen smart card, replay attacks	V2V, V2I
Ismaila et al. [55]	Elliptic curves	Privacy preservation, autonomy, non-repudiation	signature forgery, replay, man-in-the-middle, impersonation, modification attacks	V2X, IoV
Yang et al. [57]	Elliptic curves	Unforgeability, traceability and anonymity	Key exposure, coalition attacks	V2V, V2I
Zhong et al. [58]	Bilinear maps	Privacy preservation, identity privacy preservation, traceability, unforgeability	Chosen message, chosen-identity, public key replacement, replay attacks	V2I
Zhao et al. [59]	Bilinear pairing	Unforgeability, traceability, anonymity, privacy	Message forgery attack	V2I
Tomar et al. [63]	Elliptic curves, blockchain	Message authentication, identity protection, traceability, revocation,	Modification, forgery, replay and man-in-the-middle, adaptive chosen message attacks	V2V, V2I, V2E, V2X, IoV
Yang et al. [66]	Elliptic curves	Conditional privacy preservation, identity, traceability, user identity privacy, revocability	internal joint attacks, message modification, replay, impersonation attacks	V2I, IoV
Dewangan et al. [67]	Bilinear pairing	Message authentication, integrity, non-repudiation	Side-channel, message forgery, key escrow attacks	V2V, V2I
Chattaraj et al. [69]	Elliptic curves	Message authentication, integrity, non-repudiation	replay, man-in-the-middle, impersonation, privileged-insider, denial-of-service, physical vehicle capture, ephemeral secret leakage	V2V, V2I, IoV
Mei et al. [70]	Bilinear pairing	Conditional privacy preservation, anonymity, traceability	Passive KGC, replay attacks	V2V, V2I, IoV

### 2.2. Classification based on Different Authentication Methods

This section elaborates with a summary of the different authentication schemes proposed, based on certificate-less signature methods. Though there are some of the traditional methods such as mutual, cooperative, and hybrid authentication schemes, there are some of the recently developed advanced lightweight schemes, namely, smart card-based authentication, multi-factor authentication schemes, and hardware-based authentication schemes. Figure 6 represents the classification based on different authentication schemes.

#### 2.2.1. Mutual Authentication

Mutual verification or authentication is a type of digital signature verification. It is also referred to as two-factor authentication. It is an approach within which both the client and the server of the system authenticate before the data transmission. In VANETs, each node about to communicate must validate each other. The above are the works of various researchers based on mutual authentication. 

Cahyadi et al. [71] addressed the problem of information leakage on On-Board Units (OBUs) during information dissemination. The approach utilized a Tate pairing operation, thereby mitigating replay, man-in-the-middle, impersonation, and user location privacy attacks. The proposed scheme is substantially more efficient than the other competitive methods. Malhi et al. [72] came up with a new certificate-less aggregation scheme to prevent the problems of certificate management and key escrows. The proposed work mitigates replay attacks. Game-based security proofs provide prominent support for the efficacy of the proposed scheme. Investigations into other attacks are yet to be addressed further. Cui et al. [73] addressed the transmission delay that highly limited the data transmission. Their proposed work employs online/offline encryption technology. The scheme can meet the necessary security requirements, including mutual authentication, anonymity, untraceability, non-deniability, and confidentiality. The proposed scheme has to be tested under real-time constraints. Yao et al. [74] found a flaw in the traditional certificate-less aggregation scheme of message authentication without identity authentication, specifically the inability to support forward secrecy. Their proposed scheme provides privacy protection through a mutual authentication process. The scheme works for both V2V and V2I communications. Though the scheme gets relatively balanced, testing has to be performed for its practicability. Dewanta et al. [75] proposed a lightweight authentication scheme in fog computing. During the handover, the authentication between the vehicles and fog nodes, and the privacy and message integrity, are extremely needed. The scheme used a One-way Hash function, an Exclusive-OR function, and a first-come-first-serve reservation and handover schemes. The fault tolerance of the scheme has to be investigated.

Fang et al. [76] addressed the issue of security during the information dissemination in the heterogeneous Internet of Things (IoTs). The scheme has come into the picture to model the IoTs and trust relationships between the different entities within a large class of heterogenous IoT devices through mutual authentication. The work used physical unclonable functions (PUFs), and Hash and Ex-OR functions. The scheme achieves anonymity while performing mutual authentication. Gope et al. [77] addressed the problems of security and privacy in ubiquitous mobile networks. Their scheme was proposed to provide robust anonymous mutual authentication in a cloud environment, with the help of smart cards, service clouds, and hash chains. The scheme is immune to DoS, forgery, lost smart cards, and open side-channel attacks. Many features need to be taken and investigated. Mbarek et al. [78] addressed the problem of radio-frequency identification (RFID) authentication. It is always possible to encounter the problem of security issues when a jamming attack occurs. The proposed scheme can be utilized in smart home surveillance systems. The efficiency and energy consumption rates have to be investigated further. Zhu et al. [79] proposed a static random-access memory-based physical unclonable function (SRAM–PUF)-based lightweight mutual authentication scheme for the IoTs. The work has been proposed to mitigate forgery and illegal access. To guarantee the authenticity of the devices and nodes under low overhead conditions and that they are resistant to attacks, SRAM–PUFs are used. Though the scheme achieves practicability, testing has to be performed under real-time constraints. Xi et al. [80] addressed the issues of user privacy and efficiency. A zero-knowledge proof (ZKP)-based anonymous mutual authentication scheme has been proposed for use in Internet of Vehicles (IoVs). The proposed work utilized zero-knowledge proofs (ZKPs) based on a Fujisaki–Okamoto commitment algorithm and fast reconnection procedures. Their proposed scheme is resistant to replay attacks. The computational overhead needs to be reduced further. Noura et al. [81] proposed a lightweight mutual multi-factor authentication scheme for the IoTs. The proposed scheme supports both V2V and V2I communications. The work uses PUFs and entity-based fingerprints for encryption, etc. The scheme mitigates replay and MiTM attacks; it is highly immune to authentication attacks. However, the scheme has to be tested for real-time practical constraints. Alsahlani et al. [82] came up with a solution to the data transmission security and privacy issue during COVID-19 with the help of lightweight mutual multi-factor authentication, which was utilized for an IoT-based cloud environment. The scheme is highly suited for V2V and V2I communications. Alsahwish et al. [83] addressed the problem of security through mutual authentication in the Internet of Things (IoTs). However, end-to-end authentication is still a challenge. Fan et al. [84] addressed the problem of security and privacy. The authors proposed a cloud-based lightweight secure RFID mutual authentication protocol for the IoTs to achieve message authentication. The scheme has to be tested in an unsecured environment. Jan et al. [85] proposed a lightweight mutual authentication scheme to ensure data security in the IoTs for smart home surveillance. The system has been tested and utilized for time-critical applications. Table 5 provides the classification of mutual authentication schemes along with the requirements, methodology, and scenarios they suit.

#### 2.2.2. Cooperative Authentication

Cooperative authentication increases the signature verification cost of the same message through the cooperation between vehicles. The two ways of cooperative authentication are: single-message cooperative authentication and parallel cooperative authentication. Single message cooperative authentication enables immediate, low-delay message authentication. Multi-vehicle parallel cooperative authentication effectively eliminates the concerns regarding the high computation cost of bilinear pairing, thus breaking the application limitation of bilinear-pairing-based signatures in resource-constrained scenarios. Many works were carried out by various researchers concerning cooperative authentication. Yang et al. [86] addressed the problem of message authentication delay in batch verification. The most prevalent problem is that the increase in authentication delay increases the message drop rate. Their proposed work mitigates forgery attacks. The work has to be tested for real-time scenarios. Tao Jing et al. [87] proposed an efficient anonymous batch authentication (EABA) in which the authors represented the problems of information loss and the workload of RSUs. Here, the RSU workload has to be reduced. The proposed scheme has to resist against message forgery attacks. The scheme has certain limitations, as follows: 1. Only a numerical analysis has been presented. 2. Cooperative authentication in the case of urban scenarios with road cuts, intersections, and blind-spots is not considered. Xie et al. [88] explained that vehicles are unwilling to participate in message authentication or false authentication because cooperative authentication results in a loss of privacy and resource consumption. The work involves reputation mechanisms and message sequence optimization algorithm approaches. The scheme avoids message forgery attacks. Their scheme has to be tested for real-time conditions. Lin et al. [89] addressed the issue of authentication overhead on individual vehicles and shortened the authentication delay. This scheme maximally eliminates redundant authentication efforts on the same message from different vehicles. A limitation is that the trusted authority is indirectly involved in the authentication process that increases the computation overhead. Hyo Jin Jo [90] addressed the problem of message authentication. Vehicle location tracking is one of the major problems that is also addressed by them. Its major advantage is that it does not require mode synchronization between cooperative and non-cooperative authentication. The process of updation of missing messages needs to be taken care of in further work. Table 6 depicts the classification of cooperative authentication schemes along with their requirements, methodology, and scenarios.

**Table 5 sensors-23-02682-t005:** Classification of mutual-authentication-based certificate-less aggregate signature schemes.

Schemes	Methodology	Requirement	Attack Resistivity	Scenario
Cahyadi et al. [71]	Bilinear maps	Pseudonymity, identity privacy preserving, mutual authentication, message authentication, non-repudiation, traceability, unlinkability, user location privacy	Replay, man-in-the-middle, masquerade, impersonation attacks	V2V, V2I
Malhi et al. [72]	Bilinear maps	Integrity, authenticity, privacy, and non-repudiation	Replay, forgery, adaptive chosen message, signature forgery attacks	V2V, V2I
Cui et al. [73]	Elliptic curves, online/offline encryption technology, signcryption	Mutual authentication, anonymity, untraceability, non-deniability	Eavesdropping, false message, message tampering, replay, denial-of-service attacks	V2V, V2I, IoV
Yao et al. [74]	Elliptic curves	Privacy preservation, forward secrecy, message and identity Authentication	Impersonation, forgery attacks, replay, known key secrecy attacks	V2V, V2I
Dewanta et al. [75]	Elliptic curves	Mutual authentication	Replay, man-in-the-middle, arbitrary guessing attack, impersonation attack, stolen OBU/vehicle attack, replay attack, sniffing attacks	V2V, V2I
Xi et al. [80]	Zero-knowledge proofs, elliptic curves, Fujisaki–Okamoto commitment algorithm	Anonymity, mutual authenticity, unlinkability, traceability, forward secrecy	Replay attack	IoV, V2V, V2I

**Table 6 sensors-23-02682-t006:** Classification of cooperative-authentication-based certificate-less aggregate signature schemes.

Scheme	Methodology	Requirement	Attack Resistivity	Scenario
Yang et al. [86]	Bilinear pairing	Message authentication delay	Forgery, denial-of-service, active, pseudonym, replay attacks	V2V
Tao Jing et al. [87]	Elliptic curves, message classification algorithm, auction-based cooperative authentication	Information loss, identity privacy preservation, traceability	Malicious tracking attacks	V2V, V2I
Xie et al. [88]	Group member registration protocol, dynamic reputation management, sequence optimization algorithm	False authentication, loss of privacy, packet loss, missing detection ratio	Malicious attack	V2V, V2I
Lin et al. [89]	bilinear pairing, signcryption, evidence-token approach	Message authentication delay, workload	Free riding attacks	V2V, V2I
Hyo Jin jo et al. [90]	Elliptic curves	Vehicle location privacy, anonymity, authentication, revocation, conditional privacy preservation, forward secrecy	Colluding attacks	V2V, V2I

#### 2.2.3. Hybrid Authentication

Authentications can also be obtained through combinations of batch, group, mutual, cooperative, hybrid, and other verification mechanisms [91]. This combination of different authentication techniques is referred to as hybrid authentication. The following are the works carried out by various authors pertaining to the combination of different verification schemes. Tan et al. [92] proposed a certificate-less authentication scheme for efficient road message dissemination in vehicular ad hoc networks. Their proposed work combines both group and mutual authentication mechanisms, thereby mitigating forgery and replay attacks. The scheme has to improve a lot with respect to the handling of other types of attacks. Altaf et al. [93] proposed a privacy-preserving localized hybrid authentication scheme that is more efficient for large-scale VANETs, thereby eliminating the problems of central dependence on trusted authorities, frequent updates, and latency. Their proposed work resists open side-channel attacks. The verification costs have to be reduced further. Xue et al. [94] came up with a distributed authentication scheme by addressing the problems of central dependency on trusted authorities, roaming fraud, and single point failure. Their proposed work is based on blockchain-based technology that can be utilized for mobile vehicular networks. It offers roaming services efficiently and it is resistant to forgery, modification, replay, and man-in-the-middle attacks. However, authentication delays that need to be addressed still persist. Vijayakumar et al. [95] came up with a privacy-preserving mutual and batch authentication mechanism for the Internet of Things in a vehicular environment. The scheme can be utilized for safety-critical applications. The limitations are that the computation cost has to be reduced further, and that the key distribution and key management strategies have to be enhanced further. Table 7 depicts the classification of various works on hybrid authentication schemes along with their methodologies, challenges, and limitations.

**Table 7 sensors-23-02682-t007:** Classification of hybrid-authentication-based certificate-less aggregate signature schemes.

Schemes	Methodology	Requirement	Attack Resistivity	Scenario
Tan et al. [92]	Bilinear pairing, road message priority management the and dissemination Mechanism	Road message dissemination, unforgeability, forward secrecy, session key establishment, mutual authentication	Adaptive chosen message, replay attacks	V2V, V2I
Altaf et al. [93]	Gap Diffie–Hellman group bilinear pairing	Leakage resilience, low latency, localization, conditional anonymity, privacy preservation, role separation	RSU workload, open side-channel, key exposure, forgery, impersonation attacks	V2V, V2I
Xue et al. [93]	Blockchain, roaming, elliptic curves, authentication protocol, Bloom filter, undeniable billing scheme	mutual authentication, forward/backward secrecy, revocation, unforgeability, undeniability	Modification, replay, man-in-the-middle attacks	V2V, V2I, V2P
Vijayakumar et al. [95]	Bilinear maps	Anonymity, identity privacy preservation	Forgery attack	V2V

#### 2.2.4. Smart-Card-Based Authentication Schemes

The smart-card-based authentication scheme provides a means for verifying the vehicle with its corresponding components in VANETs, such as trusted authority and Road-side Units, via the use of a physical card which is accompanied by a smart card reader and has software installed at the Road-side Units or at the trusted authority. Researchers have made a significant research contribution since this method involves the use of commercially feasible, available, cheap, low-cost, powerful, secure, and efficient smart cards. This consists of the consideration of two-factor, three-factor, or multi-factor considerations while performing authentication, in order to ensure the security and the privacy of VANETs.

***i. Two-Factor Authentication schemes:*** Lu et al. [96] proposed an anonymous two-factor-based key agreement scheme for a session initiation protocol, utilizing elliptic curves. Lu et al. proposed in order to overcome the pitfalls of the other smart-card-based authentication protocols. The proposed scheme is efficient against user identity tracing, stolen smart card, key compromise, masquerading, and off-line password guessing attacks. Zhou et al. [97] proposed a bitcoin-based simplified payment verification protocol. The proposed protocol is efficient and secure, which incurs very low computation and communication costs when compared with its counterparts. Zhang et al. [98] proposed a two-factor authentication protocol that utilized elliptic curves to ensure user anonymity. The proposed work has been more efficient in alleviating various security attacks. Table 8 highlights the classification of various smart-card-based two-factor authentication schemes

***ii. Three-Factor Authentication Schemes:*** Wang et al. [99] proposed a three-factor authentication protocol that utilized elliptic curves and a fuzzy extractor algorithm, in order to ensure privacy preservation and traceability. The proposed protocol has been efficient against replay, off-line password guessing, impersonation, stolen smart card, man-in-the-middle, and insider attacks. It is more suitable for both V2V and V2I Communications. Wazid et al. [100] proposed a secure three-factor authentication scheme for energy-based smart grid environments. The proposed work utilized smart cards. The proposed work achieves a better trade-off between security and privacy than its counterparts. Xu et al. [101] came up with a three-factor authentication scheme for VANETs in order to ensure privacy preservation and user anonymity. The algorithm suffers a serious drawback in central dependency. When trusted authority gets affected, it is impossible for it to function, which is a disadvantage. Duan et al. [102] also came up with a three-factor authentication scheme in order to ensure security and privacy. It also suffers from a central dependency on the trusted servers, which poses a disadvantage. Table 9 represents the classification of various smart-card-based three-factor authentication schemes.

***iii. Multi-factor Authentication Schemes:*** Hegde et al. [103] proposed a multi-factor zero-knowledge proof-based authentication scheme in order to ensure full privacy preservation. Since the proposed scheme is based on biometrics, it is highly secure. Kebande et al. [104] came up with a blockchain-enabled multi-factor authentication model for a cloud-enabled Internet of vehicles environment. The proposed work utilized an embedded probabilistic polynomial time algorithm, which proves its efficiency by considerably overcoming the security threats. Table 10 represents the classification of various types of multi-factor-based authentication schemes.

#### 2.2.5. Hardware-Based Authentication

***i. Physical Unclonable Function based Hardware Authentication Scheme:*** Umar et al. [105] proposed a secure-identity-based anonymous inter-vehicular authentication protocol that used physically unclonable functions in VANETs to achieve privacy preservation. Though the proposed protocol is efficient, it still suffers from a high communication cost of 2976 bits. Significantly, the proposed work achieves computational efficiency. Therefore, the authentication delay will be high, which makes it infeasible for high-speed dynamic varying environments like VANETs. Othman et al. [106] addressed the problem of secure message transmission in vehicular communications. The security issue was addressed by utilizing a physically secure privacy-preserving message authentication protocol that utilized a physical unclonable function (PUF). The proposed protocol is efficient for security and privacy against passive and active attacks, even under memory leakage. The proposed protocol is efficient in achieving computation efficiency, but it still suffers from high communication costs, which need to be addressed. Table 11 represents the classification of the various types of physically unclonable function-based hardware authentication schemes. ***ii. RFID based Hardware Authentication Scheme:*** Akram et al. [107] proposed a secure and energy efficient RFID-based authentication scheme that utilized vehicular clouds. The proposed work achieves a high computation efficiency by reducing unwanted overheads, but it still suffers from high communication costs, which makes it bad. This will increase the communication delay in dynamic varying environments like VANETs. Table 12 represents the classification of the various types of RFID-based hardware authentication schemes. 

#### 2.2.6. Other Authentication Schemes

Guanquan et al. [108] addressed the problem of high computation overheads due to certificate management, key escrow problems, and central dependence on trusted authorities. A signature-enhanced certificate-less signature authentication protocol can be utilized for Vehicle-to-Infrastructure communications. The proposed work mitigates information injection attacks. This work follows elliptic curve cryptography (ECC)-based RFID authentication. The group management of vehicles in urban scenarios needs to be investigated further. Cui et al. [109] came up with an authentication scheme to reduce the problem of computation and communication overheads. Since vehicles need to store certificate revocation lists (CRLs), they are wasting storage and communication resources. The scheme uses pseudonym-based batch authentication. A limitation is that the scheme uses expensive operation-like bilinear pairings, which poses a disadvantage. Liang et al. [110] proved that certificate-less aggregation schemes are insecure and are prone to the forgery of signatures; thus, they proposed a new scheme. Their scheme involves conditional privacy preservation for the Internet of vehicles, which can be utilized for both V2V and V2I communications. Validations and security improvements are to be considered for future work. Ullah et al. [111] came up with an authentication scheme to ensure the legibility and confidentiality of message during transmission. Liao et al. [112] addressed the problems of authentication and privacy. Their scheme utilized an ECC-based RFID authentication scheme. The practicability of the proposed scheme is to be investigated further. Pino Caballero-Gil et al. [113,114] addressed the problem of the revocation of vehicles. The performance of the revocation of vehicles has to be improved. The scheme utilized tree-based revocation, which used ID-based authentication. The proposed scheme is resistant to side-channel leakage attacks, and many security constraints are yet to be addressed. Hathal et al. [115] proposed a certificate-less lightweight authentication scheme. The scheme is a broadcast authentication protocol that is well-suited for V2V and V2I communications. Their scheme utilized symmetric authentication to achieve computation and communication efficiency. The major problem is that the execution time takes the verification of the vehicles and the signature based on the trajectory information of the vehicles, which is impossible in cases of scarce environments. Table 13 represents the classification of the various types of other types of authentication schemes in VANETs.

***i. Signcryption:*** Ullah et al. [111] addressed the problem of ensuring the credibility of the transmitted data on an open wireless channel to provide receiver anonymity (only the sender knows the identity of the receiver). The authors proposed an encryption scheme with a hyperelliptic curve for an Internet of vehicles scenario that can be utilized for V2V and V2I communications. The limitations are that the scheme avoids receiver anonymity in open wireless channels.

***ii. Homomorphic Encryption:*** Lv et al. [116] addressed the problem of computational efficiency, which is directly proportional to that of the speed and coverage of the Road-side Units, which has an adverse effect on the safety of the vehicles. To maintain the route of the particular vehicle without being exposed, a privacy-preserving lightweight authentication scheme, utilized for Vehicle-to-Infrastructure communication. The proposed scheme utilized Moore curves, which are a protected via a homomorphic encryption technique. The scheme achieves fast authentication, which is suitable for dynamic mobile environments like VANETs. Though the proposed algorithm achieves computational efficiency, the increase in the number of Road-side Units increases the overhead considerably, which remains a drawback.

***iii. Genetic Algorithm:*** Ghadeer et al. [117] addressed the problem of privacy-aware secure routing, which employs an elliptic-curve-based genetic algorithm that achieves a good authentication performance. The proposed routing strategy based its concept on the optimal distribution of Road-side Units. The proposed algorithm achieves a good performance in terms of energy efficiency, packet delivery ratio, overhead, and packet loss. The proposed algorithm effectively mitigates Sybil and black-hole attacks efficiently. Further contributions can be made in optimizing the distribution of Road-side Units in VANETs [113]. 

***iv. Modeling:*** Iqbal et al. [118] proposed an IoT-based formal vehicle life integration model, which is based on the fog-based Road-side Units to assist vehicular communication in cases of emergency and life-critical situations. The proposed work incorporates unified modeling language (UML), graph theory, and VDM-SL (Vienna Development Method-Specification Language), which are suitable for vehicle–vehicle communication. The proposed work achieves its efficiency in detecting the flaws, thereby ensuring the security and the accuracy of the system. Boneh–Goh–Nissim (BGN) [119] a homomorphic encryption technique to ensure confidentiality but the major drawback is the computational overhead it faces. Jenefa et.al [120] came up with a survey that emphasize on various means of modeling techniques utilized to design a security scheme that will benefit the data transmission in vehicular ad hoc networks.

**Table 8 sensors-23-02682-t008:** Classification of smart card (two-factor) authentication schemes.

Schemes	Methodology	Requirement	Attack Resistivity	Scenario
Lu et al. [96]	Elliptic curves, Session Initiation Protocol (SIP), two-factor authentication	Anonymity, forward secrecy	Tracing, pre-authentication, key compromise masquerading, off-line password guessing attacks	V2V, V2I
Zhou et al. [97]	Elliptic curves, Bitcoin Simple Payment verification protocol (SPV)	User anonymity	Forgery, smart card stolen, privileged insider, off-line password guessing attacks	-
Zhang et al. [98]	Elliptic curves, two-factor remote authentication protocol	User anonymity, forward secrecy	replay, man-in-the-middle, impersonation, stolen-verifier table, offline dictionary, insider, stolen smart card, off-line password guessing, key exposure	-

**Table 9 sensors-23-02682-t009:** Classification of smart card (three-factor) authentication schemes.

Schemes	Methodology	Requirement	Attack Resistivity	Scenario
Wang et al. [99]	Elliptic curves, fuzzy extractor algorithm, three-factor authentication protocol	Privacy preservation, user anonymity, traceability, perfect forward secrecy	Replay, off-line password guessing, impersonation, stolen smart card, MitM, insider attacks	V2V, V2I
Wazid et al. [97]	Elliptic curves, bitwise XOR, three-factor user authentication scheme	Anonymity, untraceability	Password attack, replay, man-in-the-middle, privileged insider, user impersonation	IoV, Smart Grid
Xu et al. [101]	Elliptic curves, fuzzy extractor, three-factor authentication scheme	Privacy preservation, anonymity, untraceability, forward/backward secrecy	Replay, off-line password guessing, impersonation, smart card loss, MitM	V2V, V2I
Duan et al. [102]	Elliptic curves, fuzzy extractor, three-factor authentication	Anonymity, untraceability, perfect forward secrecy	Replay, off-line password guessing, impersonation, stolen smart card, MitM, insider Attacks	V2V, V2I, V2C

**Table 10 sensors-23-02682-t010:** Classification of multi-factor authentication schemes.

Schemes	Methodology	Requirement	Attack Resistivity	Scenario
Hegde et al. [103]	Multi-factor zero-knowledge proof authentication	User identity verification, privacy preservation, untraceability, location privacy	Replay, off-line password guessing, impersonation, smart card loss, man-in-the-middle, insider attacks	V2V, V2I
Kebande et al. [104]	Elliptic curves, blockchain, multi-factor authentication embedded probabilistic polynomial time algorithm (ePPTA)	Confidentiality, integrity, availability	Forking blockchain attacks	V2V, V2C, IoV

**Table 11 sensors-23-02682-t011:** Classification of hardware-physical-unclonable-function-based authentication schemes.

Schemes	Methodology	Requirement	Attack Resistivity	Scenario
Umar et al. [105]	Physically unclonable function (PUFs), XOR function	Vehicle anonymity, traceability, forward secrecy	Physical attack, impersonation, desynchronization, vehicle/RSU	V2I
Othman et al. [106]	Physically unclonable function (PUFs), pairwise temporal secret keys (PTKs), polynomial-based encryption, fuzzy extractor	Privacy preservation, message authentication, integrity, physical protection, confidentiality, untraceability	Known passive, active, DoS, collusion, impersonation, replay, man-in-the-middle attacks	V2V

**Table 12 sensors-23-02682-t012:** Classification of hardware-RFID-tag-based authentication schemes.

Schemes	Methodology	Requirement	Attack Resistivity	Scenario
Akram et al. [107]	RFID authentication scheme, chaotic map	Tag anonymity, untraceability	Impersonation, ephemeral secret leakage (ESL), man-in-the-middle, synchronization attack	V2V, V2C, V2D, V2I

**Table 13 sensors-23-02682-t013:** Classification of other authentication schemes.

Schemes	Methodology	Requirement	Attack Resistivity	Scenario
Guanquan et al. [108]	Elliptic curves	Message authentication, privacy preservation, traceability	Information injection, signature tampering attack	V2I
Cui et al. [109]	Elliptic curves	message integrity, non-repudiation, identity privacy preservation, traceability, revocation	Chosen message, signature forgery, man-in-the-middle, replay attacks	V2V, V2I
Ullah et al. [111]	Hyperelliptic curves (HEC), signcryption	confidentiality, unforgeability, receiver anonymity	Signature forgery attack	V2V, V2I, V2X, IoV
Hathal et al. [115]	Elliptic curves, HMAC, authentication tokens, Schnorr signature, TESLA broadcast authentication protocol, Chinese remainder theorem (CRT)	Authentication, integrity, non-repudiation, confidentiality	Impersonation, message modification, replay, DoS attacks	V2V, V2I, IoV
Lv et al. [116]	Moore curves, homomorphic encryption	Privacy preservation, identity anonymity, route plan privacy	Location privacy attacks	V2I
Ghadeer et al. [117]	Elliptic curves, genetic algorithm	Security, privacy	Sybil, black-hole Attacks	V2V, V2I
Iqbal et al. [118]	IoT, graph theory, (UML) modeling, Broadcast Emergency Message Algorithm	Security, privacy	-	V2V, V2P, V2D, V2S, IoV

## 3. Performance Comparison of Different Certificate-Less Aggregate Signature Schemes

In the existing literature, the major work has been focused on privacy preservation (conditional), anonymity, and privacy, thereby reducing the computation and communication costs, and the delay involved in authenticating the messages prior to their transmission. The existing literature utilized the communication cost calculated in bytes and the computation cost calculated in milliseconds per message as parameters, in order to evaluate the performance of the proposed authentication schemes. Some of the important benchmarks have been chosen and are analyzed for the comparison, and the insights have been presented.

Table 14 and Table 15 show the comparison of the signature verification schemes (batch vs group) based on the attacks mitigated and their performance aspects. From these tables, it is inferred that batch-based signature verification schemes are highly resistant to signature forgery, replay, modification, impersonation, man-in-the-middle, stolen-verifier tables, denial-of-service, and coalition attacks. In order to achieve a high efficiency in terms of computation, group-based elliptic curve cryptography provides high privacy and security towards various kinds of attacks. It is also evident that group-based signature verification exhibits high communication costs, since a single global public key occupies its place in determining its security and privacy aspects. Table 16 and Table 17 show the comparison of the authentication schemes (mutual, cooperative, and hybrid) based on the attacks mitigated and their performance aspects. From these tables, it is apparent that mutual authentication is inefficient in addressing attacks such as message modification, malicious, open side-channel, free riding, key exposure, collusion, and tracking attacks. It is also evident that cooperative authentication schemes are efficient against forgery, malicious, DoS, free riding, collusion, and tracking attacks. The hybrid authentication schemes combine either mutual and batch or mutual and group-based authentication schemes, which makes them highly efficient, more so than all the other authentication methods. Also, it can be seen from Table 14, Table 15, Table 16 and Table 17 that mutual authentication is used when smart cards are preferred. In the case of hybrid authentication, communication costs are high when compared with mutual and cooperative authentication schemes. From Table 18, it is inferred that the smart-card-based two-factor authentication schemes exhibit higher performance when countering attacks, while three-factor does not. When the number of consideration factors increases, the increase in the central dependence on trusted authorities also increases, this increases the communication and computation costs. From Table 19, it is apparent that two-factor authentication schemes achieve high computation efficiency when compared with their counterparts. Table 20 and Table 21 show the comparison of hardware-based authentication schemes in terms of the attacks mitigated and their performance aspects, either by using physical unclonable functions (PUFs) or radio-frequency identification (RFID) tags. From Table 17, Table 18, Table 19 and Table 20, it is evident that the physical-unclonable-function-based hardware authentication scheme exhibits a high performance in countering most of the attacks, while the radio frequency identification-based hardware authentication scheme exhibits inefficiency against achieving anonymity, traceability, replay, man-in-the-middle, impersonation, and physical and forward secrecy attacks. Among these, two-factor physical-unclonable-function-based authentication schemes also eliminate the need for relying on tamper-proof devices where physical capture attacks are possible. Both of these authentication schemes achieve low computation but high communication costs. Hence, physical-unclonable-function-based hardware authentication schemes are highly preferred, because when the radio-frequency identification tag gets lost or damaged, it would be impossible to achieve the desired authentication. 

## 4. Conclusions

Intelligent Transportation Systems act as a backbone for any smart city. Vehicular ad hoc networks are the underneath infrastructure that provide the functioning of Intelligent Transportation Systems. Since they operate via open wireless communication channels, data have to be secured and protected prior to their transmission. Therefore, in this paper, a survey based on signature verification schemes (batch vs. group) and different authentication methods has been classified. The schemes are identified based on their security requirements, the attacks mitigated, and the scenario they suit. This has been depicted in Section 2. To further evaluate the study, a comparison of the attacks encountered by the various schemes and a performance comparison based on the computation and communication costs, obtained by selecting the chosen benchmarks, has been depicted in Section 3. Further insight has been provided into the schemes and their suitability for different scenarios is presented. It is apparent that the combination of certificate-less procedures has proved its effectiveness; still, there is a need to reduce the computation and the communication overhead incurred. Hence, the designing of lightweight certificate-less authentication or aggregation schemes is needed. The use of invalid signature schemes needs to be investigated in the near future. Due to high mobility, limited computation resources, and shorter connectivity, it is much better to assume that research is currently focusing on lightweight certificate-less schemes, rather than invalid signature schemes. Finally, the proliferation of artificial-intelligence-based autonomous cars and vehicles will accelerate this arena in the future. Thus, VANETs-based authentication schemes need to be integrated into artificial intelligence and machine learning smart grid systems.

## Figures and Tables

**Figure 1 sensors-23-02682-f001:**
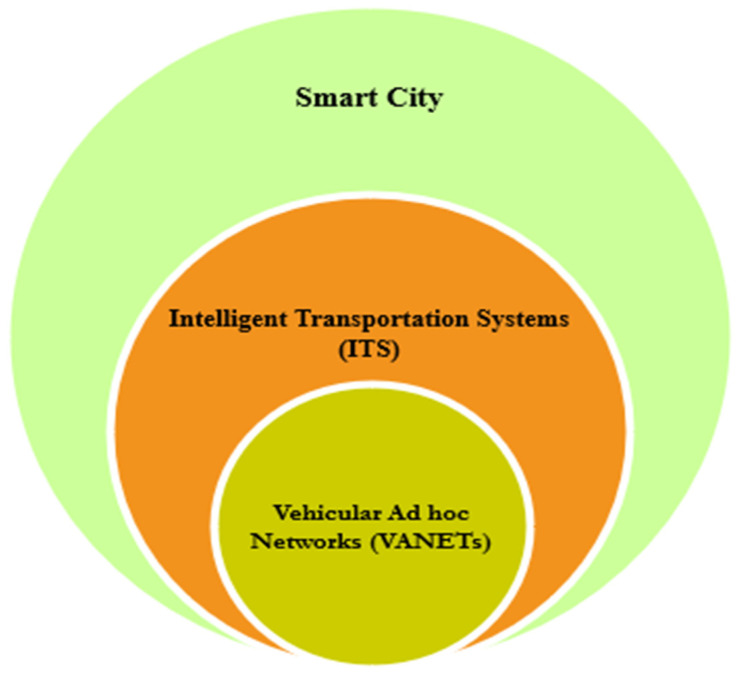
Components of a Smart City.

**Figure 2 sensors-23-02682-f002:**
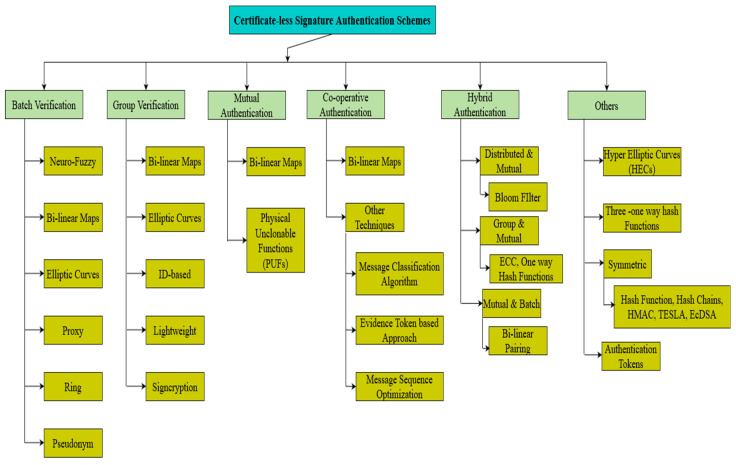
Classification of certificate-less authentication schemes (CLAS).

**Figure 3 sensors-23-02682-f003:**
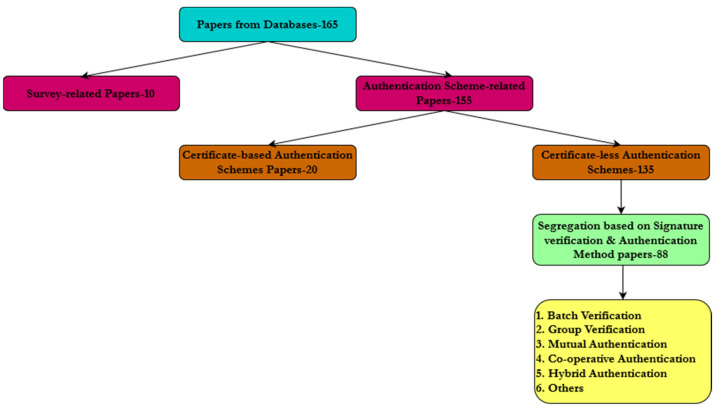
Art of collection of the literature.

**Figure 4 sensors-23-02682-f004:**
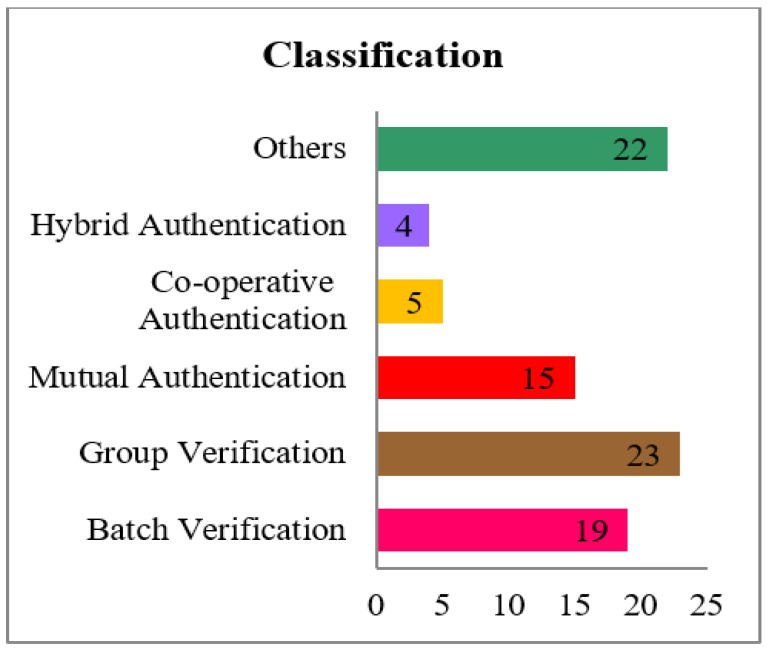
Segregation of authentication schemes.

**Figure 5 sensors-23-02682-f005:**
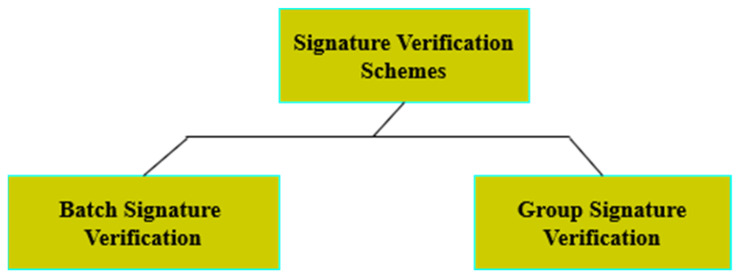
Classification based on signature verification.

**Figure 6 sensors-23-02682-f006:**
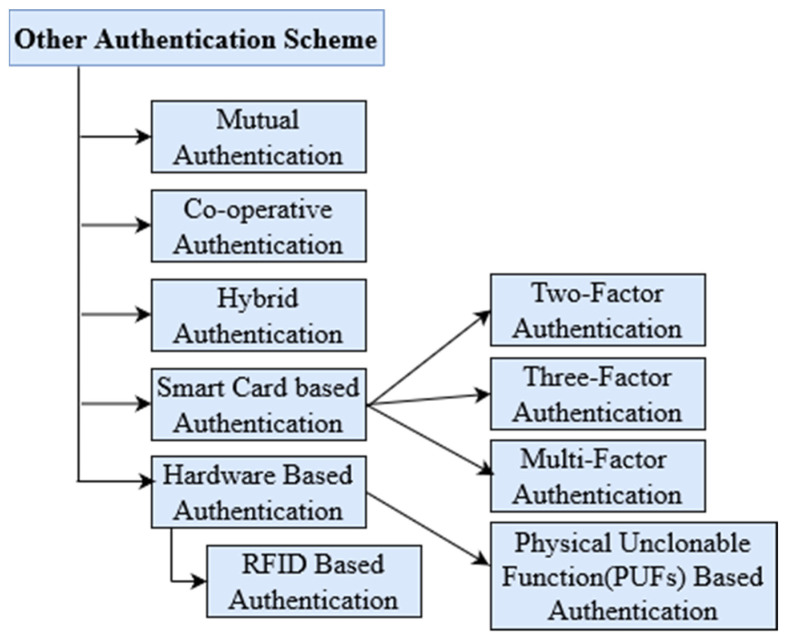
Other Types of Authentication Schemes.

**Table 14 sensors-23-02682-t014:** Comparison based on attacks mitigated by batch vs group verification schemes.

Attacks	Batch	Group
Signature forgery	Yes	Yes
Replay	Yes	Yes
Message modification	Yes	Yes
Impersonation	Yes	Yes
Man-in-the-middle	Yes	Yes
Stolen-verifier table	Yes	No
Stolen smart card	No	Yes
Denial-of-service	Yes	Yes
Anomaly	No	Yes
Coalition	Yes	Yes
Adaptive chosen message	No	Yes
Key exposure	No	Yes
Node replication	No	Yes
Side-channel	No	Yes
Vehicle capture	No	Yes
Ephemeral secret leakage	No	Yes

**Table 15 sensors-23-02682-t015:** Performance comparison of batch verification vs group signature verification schemes based on communication and communication costs.

Batch Verification			Group Verification		
Schemes	Computation Cost	Communication Cost	Schemes	Computation Cost	Communication Cost
Guo et al. [26]	6.692 ms	184 bytes	Ali et al. [45]	5.9271 ms	536 bytes
Zhou et al. [33]	1.7737 ms	208 bytes	Ming et al. [48]	3.3240 ms	128 bytes
Sunday et al. [36]	1.3762 ms	148 bytes	Tomar et al. [59]	0.874 ms	224 bytes
Zhong et al. [42]	10.131 ms	192 bytes	Yang et al. [62]	0.6650 ms	480 bytes
Zhang et al. [46]	1.3297 ms	84 bytes			

**Table 16 sensors-23-02682-t016:** Comparison based on attacks mitigated by mutual, cooperative, and hybrid authentication schemes.

Attacks	Mutual	Cooperative	Hybrid
Forgery	Yes	Yes	Yes
Replay	Yes	No	Yes
Message modification	No	No	Yes
Message tampering	Yes	No	No
Impersonation	Yes	No	Yes
Masquerading	Yes	No	No
Man-in-the-middle	Yes	No	Yes
Adaptive chosen message	Yes	No	Yes
Denial-of-service	Yes	Yes	Yes
Key secrecy	Yes	No	Yes
Malicious	No	Yes	No
Stolen OBU/vehicle	Yes	No	No
Arbitrary guessing	Yes	No	No
Open side-channel	No	No	Yes
Key exposure	No	No	Yes
Free riding	No	Yes	Yes
Collusion	No	Yes	Yes
Tracking	No	Yes	No

**Table 17 sensors-23-02682-t017:** Performance comparison of mutual, cooperative, and hybrid authentication schemes based on computation and communication costs.

Mutual Authentication			Hybrid Authentication			Cooperative Authentication	
Schemes	Computation Cost (ms)	Communication Cost (Bytes)	Schemes	Computation Cost (ms)	Communication Cost (Bytes)	Schemes	Authentication Delay
Cahyadi et al. [71]	10.41	583	Xue et al. [90]	6.884	453	Yang et al. [86]	3
Malhi et al. [72]	10.8	727	Altaf et al. [91]	12.6	294	Tao Jing et al. [87]	0.2
Cui et al. [73]	40.9	100	Vijayakumar et al. [91]	5.7	1435	Xie et al. [88]	0.2
Yao et al. [74]	36.8	168				Hyo Jin Jo et al. [90]	0.2
Dewanta et al. [75]	9	896				Yang et al. [86]	3

**Table 18 sensors-23-02682-t018:** Comparison based on attacks mitigated by smart-card-based (two vs three vs multi-factor) schemes.

Attacks	Two-Factor	Three-Factor
Replay	Yes	Yes
Man-in-the-middle	Yes	Yes
Impersonation	Yes	Yes
Stolen-verifier table	Yes	No
Offline dictionary	Yes	No
Insider	Yes	Yes
Log-in	Yes	No
Password disclosure	Yes	No
Session key	Yes	No
Key exposure	Yes	No
Perfect forward secrecy	Yes	Yes
Mutual authentication	Yes	No
Password	Yes	Yes
User anonymity	Yes	Yes
Forgery	Yes	No
Stolen smart card	Yes	Yes
Tracing	Yes	Yes
Ephemeral secret leakage	No	Yes
Anonymity	No	Yes

**Table 19 sensors-23-02682-t019:** Performance comparison of smart-card-based (two vs three vs multi-factor) schemes.

Two-Factor Authentication			Three-Factor Authentication		
Schemes	Computation Cost	Communication Cost	Schemes	Computation Cost	Communication Cost
Zhou et al. [105]	0.3117 ms	2210 bytes	Wang et al. [107]	129.022 ms	2144 bits
			Wazid et al. [111]	3.59 ms	1536 bits
			Xu et al. [112]	0.9448 ms	1088 bits
			Duan et al. [118]	2.65 ms	2560 bits

**Table 20 sensors-23-02682-t020:** Comparison based on attacks mitigated by hardware-based (PUFs vs hardware) schemes.

Attacks	PUFs	Hardware
Physical attack resilience	Yes	No
Vehicle anonymity	Yes	No
Vehicle traceability	Yes	No
Mutual authentication	Yes	No
Desynchronization attacks	Yes	Yes
Impersonation attacks	Yes	No
Perfect forward secrecy	Yes	No
Cloning and physical attacks	Yes	Yes
Tag anonymity	No	Yes
Untraceability	No	Yes
Server impersonation attacks	No	Yes
Ephemeral secret leakage attack	No	Yes
Man-in-the-middle attacks	No	Yes

**Table 21 sensors-23-02682-t021:** Performance comparison of hardware-based (PUFs vs RFID) schemes.

Physical Unclonable Function Authentication			RFID Authentication		
Schemes	Computation Cost	Communication Cost	Schemes	Computation Cost	Communication Cost
Umar et al. [108]	3.7833 ms	2976 bits	Akram et al. [117]	4.222 ms	2496 bits
Othman et al. [110]	32.84 ms	3360 bits			

## Data Availability

There are no data available for this manuscript.

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
