# Peer review of "A Comprehensive Survey on Certificate-Less Authentication Schemes for Vehicular Ad hoc Networks in Intelligent Transportation Systems"

_sensors, 2023, doi:10.3390/s23052682_

Round 1

Reviewer 1 Report

1.      Authors may revise the abstract to elaborate more on the problem statement, findings, and contributions and compare their results with the existing approaches.

2.      How it contributes to the literature, in the second last paragraph of the introduction, the provided details are not sufficient

3.      Authors may elaborate on the novelty of their work, and how this is significant over the existing approaches.

4.      Overall, the paper presentation requires improvement.

5.      A few of the figures are not clearly readable such as Figure 1

6.      Thorough proofreading is recommended.

7.      A few of the references are missing some information, may complete them critically

8.      The conclusion is not clear and needs revision and clarity and alignment with the abstract and title

9.      Provided references are better enough. However, authors are recommended to consider more latest and related such as,

Hussain, K., Jhanjhi, N. Z., Mati-ur-Rahman, H., Hussain, J., & Islam, M. H. (2021). Using a systematic framework to critically analyze proposed smart card based two factor authentication schemes. Journal of King Saud University-Computer and Information Sciences33(4), 417-425.

Author Response

Reviewer 1:

Comment 1: Authors may revise the abstract to elaborate more on the problem statement, findings, and contributions and compare their results with the existing approaches.

Response 1: 

With respect to the reviewer’s comment 1, the abstract has been revised carefully and is elaborated on the problem statement, findings, contributions, comparison with the existing approaches.

Comment 2: How it contributes to the literature, in the second last paragraph of the introduction, the provided details are not sufficient.

Response 2:

According to the reviewer’s comment 2, the problem statement and the literature survey are matched perfectly. The details regarding the contribution of the literature survey to the problem statement has been analyzed and presented and the second last paragraph in the end of the introduction part has provided with enough details which are sufficient.

Comment 3: Authors may elaborate on the novelty of their work, and how this is significant over the existing approaches.

Response 3:

As per the reviewer’s comment 3, the novelty of the work has been specified in a precise manner at the end of the introduction section.

Comment 4: Overall, the paper presentation requires improvement.

Response 4:

As per the reviewer’s comment 4, the presentation of the paper has been improved.

Comment 5: A few of the figures are not clearly readable such as Figure 1

Response 5:

With respect to the reviewer’s comment 5, the figure 1 has been re-drawn and is visible and readable.

Comment 6: Thorough proofreading is recommended.

Response 6:

With respect to the reviewer’s comment 6, the manuscript has undergone a thorough proof reading by two authors who have made extensive research contributions.

Comment 7: A few of the references are missing some information, may complete them critically

Response 7:

With respect to the reviewer’s comment 7, the critical information of the references has been found and modified completely. The following references [9, 15, 18, 20, 34, 45, 67, 76, 84, 89, 109, 114, 116, 117, and 134] are corrected.

Comment 8: The conclusion is not clear and needs revision and clarity and alignment with the abstract and title.

Response 8:

With respect to the reviewer comment 8, the conclusion has been modified precisely and has been presented with enough clarity which has been in alignment with the abstract and title.

Comment 9: Provided references are better enough. However, authors are recommended to consider latest and related such as,

Hussain, K., Jhanjhi, N. Z., Mati-ur-Rahman, H., Hussain, J., & Islam, M. H. (2021). Using a systematic framework to critically analyze proposed smart card based two factor authentication schemes. Journal of King Saud University-Computer and Information Sciences, 33(4), 417-425.

Response 9:

As per the suggestion from the comment 9, the prescribed reference under smart card based two-factor, three-factor and multi-factor based authentication schemes are also included.

Reviewer 2 Report

The current version  of the manuscript mainly focuses on the comparison of different references in form. The comparative analysis is not very reasonable and enough.

It is more like a daybook of reading instead of a paper to be published, and is short of proper summary and analysis. 

It is recommended to classify the published works more carefully, and to analysis the performance of different  certificateless authentication schemes under a similar constraint.

Author Response

Reviewer 2:

Comment 1: The current version of the manuscript mainly focuses on the comparison of different references in form. The comparative analysis is not very reasonable and enough.

Response 1:

With respect to the reviewer comment 1, the manuscript also focuses on the advantages and limitations of the chosen references and it also highlights the parameters and goals met. The comparative analysis has been made reasonable by making changes necessarily in the manuscript which is subsequent.

Comment 2: It is more like a daybook of reading instead of a paper to be published, and is short of proper summary and analysis.

Response 2:

With respect to the reviewer comment 2, the manuscript has been modified by incorporating changing by including the summary and analysis for each authentication schemes accordingly.

Comment 3: It is recommended to classify the published works more carefully, and to analysis the performance of different certificate-less authentication schemes under a similar constraint.

Response 3:

With respect to the reviewer comment 3, the classification has been carefully reviewed and the analysis has been depicted by analyzing the performance aspects under a similar constraint.
